# FLUID FLOW MASS TRANSPORT FOR GENERATIVE NETWORKS

## ABSTRACT

Generative Adversarial Networks have been shown to be powerful tools for generating content resulting in them being intensively studied in recent years. Training these networks requires maximizing a generator loss and minimizing a discriminator loss, leading to a difficult saddle point problem that is slow and difficult to converge. Motivated by techniques in the registration of point clouds and the fluid flow formulation of mass transport, we investigate a new formulation that is based on strict minimization, without the need for the maximization. This formulation views the problem as a matching problem rather than an adversarial one, and thus allows us to quickly converge and obtain meaningful metrics in the optimization path.

## 1 INTRODUCTION

Generative Networks have been intensively studied in recent years yielding some impressive results in different fields (see for example Salimans et al. (2016); Karras et al. (2017); Brock et al. (2018); Karras et al. (2018); Zhu et al. (2017) and reference within). The most common technique used to train a generative network is by formulating the problem as a Generative Adversarial Networks (GANs). Nonetheless, GANs are notoriously difficult to train and in many cases do not converge, converge to undesirable points, or suffer from problems such as mode collapse.

Our goal here is to better understand the generative network problem and to investigate a new formulation and numerical optimization techniques that do not suffer from similar shortcomings. To be more specific, we consider a data set made up of two sets of vectors, template vectors (organized as a matrix) $\mathbf{T} = [\mathbf{T}_1, \ldots, \mathbf{T}_n] \in \mathcal{T}$ and reference vectors $\mathbf{R} = [\mathbf{R}_1, \ldots, \mathbf{R}_n] \in \mathcal{R}$. The goal of our network is to find a transformation $f(\mathbf{T}, \boldsymbol{\theta})$ that generates reference vectors, that is vectors from the space $\mathcal{R}$, from template vectors in the space $\mathcal{T}$, where $\boldsymbol{\theta}$ are parameters that control the function $f$. For simplicity, we write our generator as

$$\mathbf{T}(\boldsymbol{\theta}) = f(\mathbf{T}, \boldsymbol{\theta}) \tag{1.1}$$

Equation equation 1.1 defines a generator that depends on the template data and some unknown parameters to be learned in the training process. We will use a deep residual network to approximate the function with $\boldsymbol{\theta}$ being the weights of the network.

In order to find the parameters for the generator we need to minimize some loss that captures how well the generator works. Assume first that we have correspondence between the data points in $\mathbf{T}$ and $\mathbf{R}$, that is, we know that the vector $\mathbf{T}_j(\boldsymbol{\theta}) = \mathbf{R}_j, \forall j$. In other words, our training set consists of paired input and output vectors. In this case we can find $\boldsymbol{\theta}$ by simply minimizing the sum of squares difference (and may add some regularization term such as weight decay).

$$\mathcal{E}(\boldsymbol{\theta}) = \frac{1}{2n} \sum_j \|\mathbf{T}_j(\boldsymbol{\theta}) - \mathbf{R}_j\|^2.$$

However, such correspondence is not always available and therefore, a different approach is needed if we are to estimate the generator parameters.

The most commonly used approach to solve the lack of correspondence is Generative Adversarial Networks (GANs) and more recently, the Wasserstein GANs (WGANs) Arjovsky et al. (2017). In

these approaches one generates a discriminator network, or a critic in the case of WGANs, $c(\mathbf{r}, \boldsymbol{\eta})$ that gives a score for a vector, $\mathbf{r}$, to be in the space $\mathcal{R}$. In the original formulation $c(\cdot, \cdot)$ yields the probability and in more recent work on WGANs it yields a real number. The discriminator/critic depends on the parameters $\boldsymbol{\eta}$ to be estimated from the data in the training process. One then defines the function

$$\mathcal{J}(\boldsymbol{\eta}, \boldsymbol{\theta}) = g(c(\mathbf{R}, \boldsymbol{\eta})) + h(c(\mathbf{T}(\boldsymbol{\theta}), \boldsymbol{\eta})) \tag{1.2}$$

In the case of a simple GAN we have that

$$g(\mathbf{x}) = \log(s_m(\mathbf{x})) \quad \text{and} \quad h(\mathbf{x}) = \log(1 - s_m(\mathbf{x}))$$

where the $s_m(\cdot)$ function, is a soft max function that converts the score to a probability. In the case of WGANs a simpler expression is derived where we use the score directly setting $g$ to the identity and $h = -g$, and require some extra regularity on the score function.

Minimizing $\mathcal{J}$ with respect to $\boldsymbol{\eta}$ detects the "fake" vectors generated by the generator while maximizing $\mathcal{J}$ with respect to $\boldsymbol{\theta}$ "fools" the discriminator, thus generating vectors $\mathbf{T}(\boldsymbol{\theta})$ that are similar to vectors that are drawn from $\mathcal{R}$. Training GANs is a minimax problem where $\mathcal{J}(\boldsymbol{\eta}, \boldsymbol{\theta})$ is minimized with respect to $\boldsymbol{\eta}$ and maximize with respect to $\boldsymbol{\theta}$. Minimax problems are very difficult to solve and are typically unstable. Furthermore, the solution is based on gradient d/ascent which is known to be slow, especially when considering a saddle point problem Nocedal & Wright (1999), and this can be demonstrated by solving the following simple quadratic problem.

**Example 1.1** *Let*

$$\mathcal{J}(\eta, \theta) = \frac{1}{2}\theta^2 - \frac{1}{20}\eta^2 + 10\theta\,\eta$$

*The d/ascent algorithm used for the solution of the problem reads*

$$\theta_{k+1} = \theta_k - \mu(\theta_k + 10\eta_k) \qquad \eta_{k+1} = \eta_k - \mu(\frac{1}{10}\eta_k - 10\theta_{k+1})$$

*The convergence path of the method is plotted in Figure 1. It is evident that the algorithm takes an*

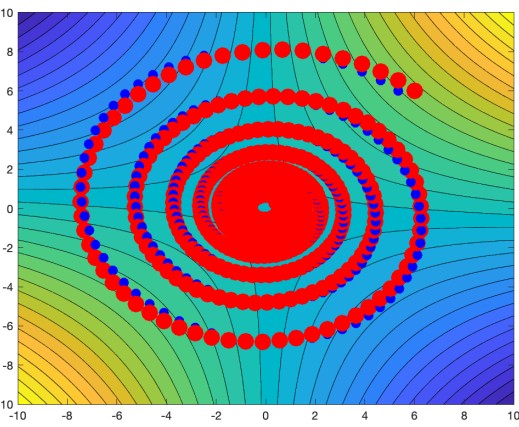

Figure 1: The convergence path for the solution of the quadratic problem in 1.1. The blue points represent the ascent step and the red the descent step. Note the circular path the algorithm takes for convergence. This path is typical when solving a saddle point problem using an a/descent method.

*inefficient path to reach its destination. This inefficient path is a known property of the method and avoiding it requires much more complex algorithms that, for example, eliminate one of the unknowns at least locally (see a discussion in Nocedal & Wright (1999)). While such approaches have been derived for problems in fluid dynamics and constrained optimization, they are much more difficult to derive for deep learning due to the non-linearity of the learning problem.*

Besides the slowly converging algorithm, the simple GAN approach has a number of known fundamental problems. It has been shown in Zhang et al. (2017) that a deep network can classify vectors

with random labels. This implies that given sufficient capacity in the classifier, it is always possible to obtain 0 loss even if $\mathbf{R} \neq \mathbf{T}(\boldsymbol{\theta}) \in \mathcal{R}$, implying that there is no real metrics to stop the process (see further discussion in Arjovsky & Bottou (2017)). This problem also leads to mode collapse, as it is possible to obtain the saddle point of the function when $\mathbf{T}(\boldsymbol{\theta}) = \mathbf{R}$, however this yields a local solution that is redundant. Therefore, it is fairly well known that the discriminator needs to be heavily regularized in order to effectively train the generator Gulrajani et al. (2017), such as by using a very small learning rate or by weight clipping in the case of WGANs. A number of improvements have been proposed for GANs and WGANs, however these techniques still involve a minimax problem that can be difficult to solve.

Another common property of all GANs is that they minimize some distance between the two vector spaces that are spanned by some probabilities. While a simple GAN is minimizing the JS-Divergence, the Wasserstein GAN minimizes the Wasserstein Distance. Minimizing the distance for probabilities makes sense when the probabilities are known exactly. However, as we discuss in the next section, such a process is not reasonable when the probabilities are estimated, that is, sampled and are therefore noisy. Since both spaces, $\mathcal{R}$ and $\mathcal{T}$ are only sampled, we only have noisy estimations of the probabilities and this has to be taken into consideration when solving the problem.

In this paper we therefore propose a different point of view that allows us to solve the problem without the need of minimax. Our point of view stems from the following observation. Assume for a moment that the vectors $\mathbf{R}$ and $\mathbf{T}(\boldsymbol{\theta})$ are in $R^2$ or $R^3$. Then, the problem described above is nothing but a registration of point clouds under a non-common transformation rule (i.e. not affine transformation). Such problem has been addressed in computer graphics and image registration for decades (see for example Eckart et al. (2018); Myronenko & Song (2010) and reference within) yielding successful software packages, numerical analysis and computational treatment. This observation is demonstrated in the following example, that we use throughout the paper.

**Example 1.2** *Assume that the space $\mathcal{T}$ is defined by vectors $\mathbf{T}_i$ that are in $R^2$ and that each vector is drawn from a simple Gaussian distribution with $0$ mean and standard deviation of $1$. To generate the space $\mathcal{R}$ we use a* **known** *generator that is a simple resnet with fully connected layers*

$$\mathbf{x}_{j+1} = \mathbf{x}_j + h\sigma(\mathbf{K}_j\mathbf{x}_j) \quad j = 1, \ldots, n \quad \mathbf{x}_1 = \mathbf{T}$$

*Here $\sigma(\cdot)$ is the $\tanh$ activation function and we choose $\mathbf{K}_j$ and save them for later use. The original points as well as points from the same distribution that are transformed using this simple resenet are plotted in Figure 2. Finding the transformation parameters (the matrices $\mathbf{K}_j$) is a kin to registering the red and blue points, when no correspondence map is given.*

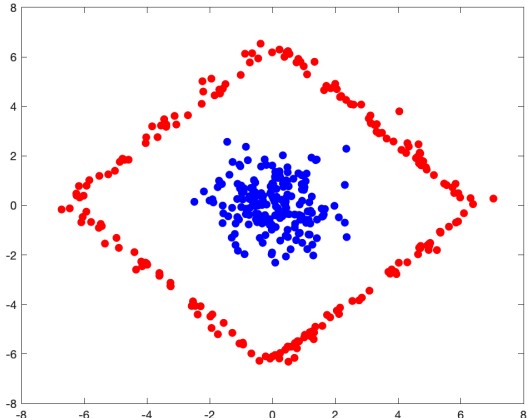

Figure 2: Template points drawn from a Gaussian random distribution (blue) are transformed using a simple ResNet to generate new reference points (red). There is no correspondence between the red/blue points. The goal of the training is to recover a transformation that move the blue points into the red ones without known correspondence.

We therefore propose to extend the ideas behind cloud point registration to higher dimensions, adapting them to Generative Models. Furthermore, recent techniques for such registration are

strongly connected to the fluid flow formulation of optimal mass transport. As we explore next, there is a strong connection between our formulation and the Wasserstein GAN. Similar connections have been proposed in Lei et al. (2019). Our approach can be viewed as a discretization of the fluid flow formulation of the optimal mass transport problem proposed in Benamou & Brenier (2003) and further discussed in Haber & Horesh (2014); Ryu et al. (2017). It has some commonalities with normalized flow generators Kingma & Dhariwal (2018) with two main differences. First, our flow become regular due to the addition of regularization that explicitly keep the flow normalized and second, and more importantly, it solves the correspondence problem between the two data sets. The fluid formulation is known to be easier to solve and less nonlinear than the standard approach and (at least in 3D) leads to better and faster algorithms. We name our algorithm Mass Transport Generative Networks (MTGN) since our algorithm is based on mass transport that tries to match distributions but not adversarial networks. Similar to the fluid flow formulation for the OMT problem, our algorithm leads to a simple minimization problem and does not require the solution of a minimax.

The rest of this paper is organized as follows. In Section 2 we lay out the foundation of the idea used to solve the problem, including discretization and numerical optimization. In Section 3 we demonstrate the idea on very simple example that help us gain insight into the method. In Section 4 we perform numerical experiments in higher dimensions and discuss how to effectively use the method and finally in Section 5 we summarize the paper and suggest future work.

## 2 GENERATIVE MODELS, MASS TRANSPORT AND CLOUD POINT REGISTRATION

In this section we discuss our approach for the solution of the problem and the connection to the registration of cloud of points and the fluid flow formulation of optimal mass transport.

### 2.1 GENERATIVE MODELS AND FLUID FLOW MASS TRANSPORT

We start by associating the spaces $\mathcal{T}$ and $\mathcal{R}$ with two probability density functions $p_T(\mathbf{x})$ and $p_R(\mathbf{x})$. The goal is to find a transformation such that the probability, $p_T$ is transformed to the probability $p_R$, minimizing some distance (see the review paper Evans (1989) for details). An $L_2$ distance leads to the Monge Kantorovich problem but it is possible to use different distances to obtain different transformations Burger et al. (2013). The computation of such a transformation has been addressed by vast amount of literature. Solution techniques range from linear programming, to the solution of the notoriously nonlinear Monge Ampre equation Evans (1989). However, in a seminal paper Benamou & Brenier (2003), it was shown that the problem can be formed as minimizing the energy of a simple flow

$$\min \quad \mathcal{E}(\mathbf{v}(\mathbf{x},t)) = \int_0^1 \int_\Omega \rho(\mathbf{x},t)|\mathbf{v}(\mathbf{x},t)|^2 \, d\mathbf{x}\, dt \tag{2.3a}$$

$$\text{s.t} \quad \rho_t + \nabla \cdot (\mathbf{v}\rho) = 0 \tag{2.3b}$$

$$\rho(0,\mathbf{x}) = p_T(\mathbf{x}) \quad \rho(1,\mathbf{x}) = p_R(\mathbf{x}) \tag{2.3c}$$

Here $\mathcal{E}$ is the total energy that depends on the velocity and density of the flow. The idea was extended in Chen et al. (2017) to solve the problem with different distances on vector spaces. The problem is also commonly solved in fields such as computational flow of fluids in porous media, where different energy and more complex transport equations are considered Sarma et al. (2007). A simple modification which we use here, is to relax the constraint $\rho(1,\mathbf{x}) = p_R(\mathbf{x})$ and to demand it holds only approximately. This formulation is better where we have noisy realizations of the distributions and a perfect fit may lead to overfitting. The formulation leads to the optimization problem

$$\min \quad \mathcal{E}(\mathbf{v}(\mathbf{x},t)) = \frac{1}{2}\int_\Omega (\rho(1,\mathbf{x}) - p_R(\mathbf{x}))^2 \, d\mathbf{x} + \alpha \int_0^1 \int_\Omega \rho(\mathbf{x},t)|\mathbf{v}(\mathbf{x},t)|^2 \, d\mathbf{x}\, dt \tag{2.4a}$$

$$\text{s.t} \quad \rho_t + \nabla \cdot (\mathbf{v}\rho) = 0 \quad \rho(0,\mathbf{x}) = p_T(\mathbf{x}) \tag{2.4b}$$

Here, the first term

$$\mathcal{M} = \frac{1}{2}\int_\Omega (\rho(1,\mathbf{x}) - p_R(\mathbf{x}))^2 \, d\mathbf{x} \tag{2.5}$$

can be viewed as a data misfit, while the second term

$$\mathcal{S} = \int_0^1 \int_\Omega \rho(\mathbf{x}, t)|\mathbf{v}(\mathbf{x}, t)|^2 \, d\mathbf{x}\delta t \tag{2.6}$$

can be thought of as regularization. Typical to all learning problems, the regularization parameter $\alpha$ needs to be chosen, usually by using some cross validation set. Such a formulation is commonly solved in applied inverse transport problems Dean & Chen (2011).

Here we see that our formulation equation 2.4a differs from both WGAN and normalized flow. It allows us to have a tradeoff between the regularity of the transformation as expressed in equation 2.6 to fitting the data as expressed in equation 2.5. This is different from both WGAN and normalized flow when such a choice is not given.

## 2.2 DISCRETIZATION OF THE REGULARIZATION TERM

The optimization problem equation 2.4 is defined in continuous space $\mathbf{x}, t$ and in order to solve it we need to discretize it. The work in Benamou & Brenier (2003); Haber & Horesh (2014) used *an Eulerian framework*, and discretize both $\mathbf{x}$ and $t$ on a grid in 2D and 3D. Such a formulation is not suitable for our problems as the dimensionality of the problem can be much higher. In this case, a *Lagrangian* approach that is based on sampling is suitable although some care must be taken if the discrete problem to be solved is faithful to the continuous one. To this end, the flow equation 2.3b is approximated by placing particles, of equal mass for now, at locations $\mathbf{x}_i = \mathbf{t}_i, i = 1, \ldots, n$ and letting them flow by the equation

$$\begin{aligned} \frac{d\mathbf{x}_i}{dt} &= \mathbf{v}(\mathbf{x}_i, t; \boldsymbol{\theta}) \quad i = 1, \ldots, n, \quad t \in [0, 1] \\ \mathbf{x}_i(0) &= \mathbf{t}_i \end{aligned} \tag{2.7}$$

Here $\mathbf{v}(\mathbf{x}_i, t; \boldsymbol{\theta})$ is the velocity field that depends on the parameters $\boldsymbol{\theta}$. If we use the forward Euler discretization in time, equation equation 2.7 is nothing but a resnet that transforms particles located in $\mathbf{x}_i = \mathbf{t}_i$ from the original distribution $p_T(\mathbf{x})$ to the final distribution $p_{T(\boldsymbol{\theta})}(\mathbf{x})$, that is sampled at points $\mathbf{x}_i(t), i = 1, \ldots, n$. It is important to stress that other discretizations in time may be more suitable for the problem. Using the point mass approximation to estimate the density, the regularization part of the energy can be approximate in a straight forward way as

$$\mathcal{S}_n(\boldsymbol{\theta}) = \frac{\delta t}{n} \sum_{ij} \|\mathbf{v}(\mathbf{x}_i, t_j; \boldsymbol{\theta})\|^2 \tag{2.8}$$

where $\delta t$ is the time interval used to discretize the ODE's equation 2.7. We see that the $L_2$ OMT energy is simply a sum of squared activations for all particles and layers. Other energies can be used as well and can sometimes lead to more regular transportation maps Burger et al. (2013).

## 2.3 DISCRETIZING THE MISFIT AND POINT OF CLOUD REGISTRATION

Estimating the first term in the objective function $\mathcal{M}$, the misfit, requires further discussion. Assume that we have used some parameters $\boldsymbol{\theta}$ and push the particles forward. The main problem is how to compare the distributions $p_{T(\boldsymbol{\theta})}$ that is sampled at points $\mathbf{T}_i(\boldsymbol{\theta}) = \mathbf{x}_i(t), i = 1, \ldots, n$ and $p_R(\mathbf{x})$ sampled at $\mathbf{R}_i, \ i = 1, \ldots, n$.

In standard Lagrangian framework one usually assumes correspondence between the particles in the different distributions, however this is not the case here. Since we have unpaired data there is no way to know which particle in $\mathbf{R}$ corresponds to a particle in $\mathbf{T}(\boldsymbol{\theta})$. This is exactly the problem solved when registering two point clouds to each other. We thus discuss the connection between our approach to point cloud registration.

One approach for measuring the difference between two point clouds is using the closest point match. This is the basis for the Iterative Closest Point (ICP) Besl & McKay (1992) algorithm that is commonly used to solve the problem. Nonetheless, the ICP algorithm tends to converge only locally and thus we turn to other algorithms that usually exhibit better properties.

Following the work Myronenko & Song (2010) we use the idea of coherent point drift for the solution of the problem. To this end, we use a Gaussian Mixture Model to evaluate the distribution of each of

the data. We define the approximations $p_n(\mathbf{x}, \mathbf{R})$ and $p_n(\mathbf{x}, \mathbf{T}(\boldsymbol{\theta}))$ to $p(\mathbf{x}, \mathbf{R})$ and $p(\mathbf{x}, \mathbf{T}(\boldsymbol{\theta}))$ as

$$p_n(\mathbf{x}, \mathbf{R}) = \sum_{i=1}^{n} \frac{1}{(2\pi\sigma^2)^{\frac{d}{2}}} \exp\left(-\frac{\|\mathbf{x} - \mathbf{R}_i\|^2}{\sigma^2}\right) \tag{2.9}$$

and

$$p_n(\mathbf{x}, \mathbf{T}(\boldsymbol{\theta})) = \sum_{i=1}^{n} \frac{1}{(2\pi\sigma^2)^{\frac{d}{2}}} \exp\left(-\frac{\|\mathbf{x} - \mathbf{T}_i(\boldsymbol{\theta})\|^2}{\sigma^2}\right) \tag{2.10}$$

The integral equation 2.5 can be now written as

$$\mathcal{M}_n(\boldsymbol{\theta}) = \frac{1}{2} \int_\Omega (p_n(\mathbf{x}, \mathbf{T}(\boldsymbol{\theta})) - p_n(\mathbf{x}, \mathbf{R}))^2 \, d\mathbf{x} \tag{2.11}$$

Finally, we approximate $\mathcal{M}_n$ by replacing $\mathbf{x}$ by the sampled points $\mathbf{T}(\boldsymbol{\theta})$ and $\mathbf{R}$ in a symmetric distance obtaining

$$\mathcal{M}_{nh}(\boldsymbol{\theta}) = \frac{1}{2}\|p_n(\mathbf{T}(\boldsymbol{\theta}), \mathbf{T}(\boldsymbol{\theta})) - p_n(\mathbf{T}(\boldsymbol{\theta}), \mathbf{R})\|^2 + \frac{1}{2}\|p_n(\mathbf{R}, \mathbf{T}(\boldsymbol{\theta})) - p_n(\mathbf{R}, \mathbf{R})\|^2 \tag{2.12}$$

To summarize, we minimize the fluid flow formulation equation 2.4 by discretizing the misfit term equation 2.5 and the regularization term equation 2.6.

## 2.4 NUMERICAL OPTIMIZATION

The optimization problem equation 2.4 can be solved using any standard optimization technique, however, there are a number of points that require special attention. First, the batch size in both $\mathbf{R}$ and $\mathbf{T}(\boldsymbol{\theta})$ cannot be too small. This is because we are trying to match probabilities that are approximated by particles. For example, using a batch of a single vector is very likely to not represent the probability density. Better approximations can be obtained by using a different approximation to the distribution, for example, by using a small number of Gaussians but this is not explored here. A second point is the choice of $\sigma$ is the estimation of the probability. When the distributions are very far it is best to pick a rather large $\sigma$. Such a choice yields a very "low resolution" approximation to the density, that is, only the low frequencies of the densities are approximated. As the fit becomes better, we decrease $\sigma$ and obtain more details in the density's surfaces. This principle is very well known in image registration Modersitzki (2004).

## 3 NUMERICAL EXPERIMENTS ON SYNTHETIC DATA

In this section we perform numerical experiments using synthetic data. The goals of these experiments are twofold. First, experimenting in 2D allows us to plot the distributions and obtain some highly needed intuition. Second, synthetic experiments allow us to quantitatively test the results as we can always compute the true correspondence for a new data point.

Returning to Example 1.2, we use the data generated with some chosen parameters $\boldsymbol{\theta}^{\text{true}}$. We train the generator to estimate $\boldsymbol{\theta}$ and obtain convergence in 8 epochs. The optimization path is plotted in Figure 3. We have also used a standard GAN Zhu et al. (2017) in order to achieve the same goal. The GAN converged much slower and to a visually less pleasing solution that can be qualitatively assessed to be of lower accuracy.

One of the advantages of synthetic experiments is that we have the "true" transformation and therefore can qualitatively validate our results. To this end, we choose a new set of random points, $\mathbf{T}^{\text{test}}$ and used them with the optimal parameters $\boldsymbol{\theta}$ to generate $\mathbf{T}^{\text{test}}(\boldsymbol{\theta})$ and its associate approximate distribution. We also generate the "true" distribution from the chosen parameters $\boldsymbol{\theta}^{\text{true}}$ by pushing $\mathbf{T}^{\text{test}}$ with the true parameters, generating $\mathbf{T}^{\text{test}}(\boldsymbol{\theta}^{\text{true}})$ We then compute the mean square error

$$E = \frac{\|\mathbf{T}^{\text{test}}(\boldsymbol{\theta}) - \mathbf{T}^{\text{test}}(\boldsymbol{\theta}^{\text{true}})\|}{\|\mathbf{T}^{\text{test}}(\boldsymbol{\theta}^{\text{true}})\|}$$

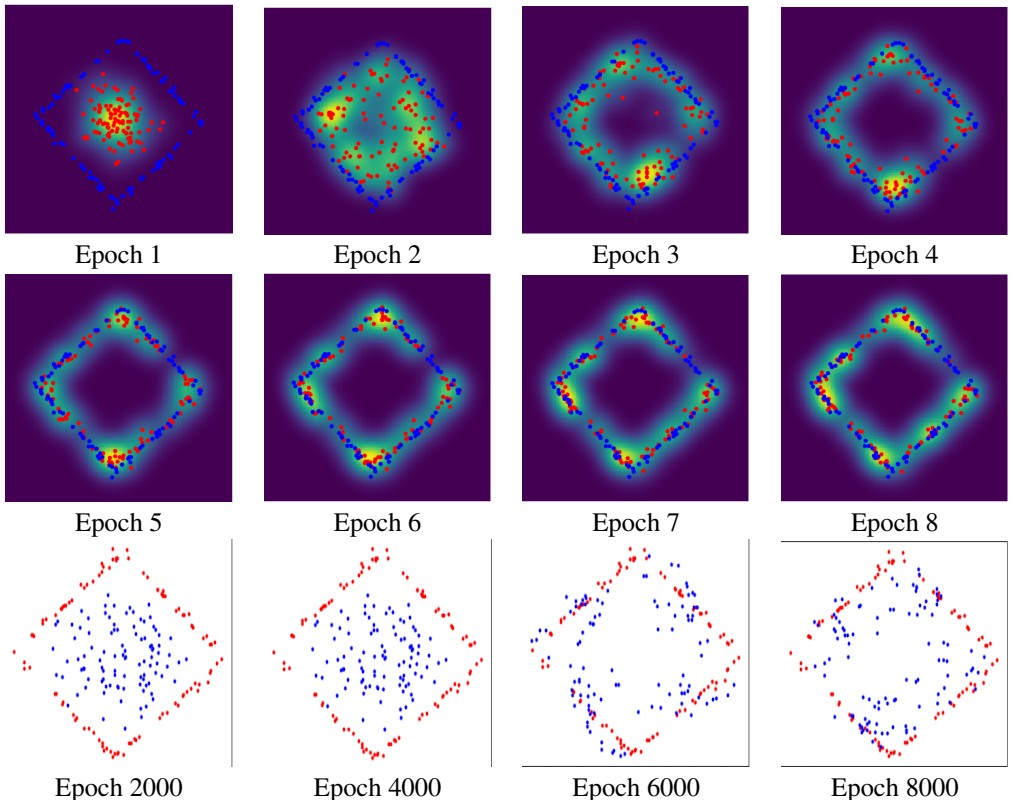

Figure 3: Path of the optimization. Top two panels: The template points are transformed into the reference space using our approach in 8 epochs. The density estimated from the red points is displayed in colour. The bottom panel is the path of optimization taken by the GAN training. It takes more than 8000 epochs and the results are clearly not worse.

For the experiment at hand we obtained an error of $E = 8 \times 10^{-2}$ with our method, while the GAN gave us an error of $E = 3.6 \times 10^{-1}$, which is substantially worse than our estimated network. This implies that the using our training the network managed to learn the transformation rather well. Unfortunately, this quantitative measure can only be obtained when the transformation is known and this is why we believe that such simple tests are important.

## 4 NUMERICAL EXPERIMENTS IN HIGHER DIMENSIONS

In order to match higher dimensional vectors and distributions we slightly modify the architecture of the problem. Rather than working on the spaces $\mathcal{T}$ and $\mathcal{R}$ directly, we use a feature extractor to obtain latent space $\mathcal{T}_\ell$ and $\mathcal{R}_\ell$ respectively. Such spaces can be formed for example by training an auto-encoder and then use the encoded space as the latent space. We then register the points in $\mathcal{T}_\ell$ to the points in $\mathcal{R}_\ell$. In the experiments we have done here we used the MNIST data set and used a simple encoder similar to Kingma & Welling (2019) to learn the latent space of $\mathcal{R}$. We then use our framework to obtain the transformation that maps a template vector sampled from a Gaussian random distribution, $\mathcal{T}_\ell$, with 0 mean and a standard deviation of 1 to the latent space $\mathcal{R}_\ell$. In our experiments the size of the latent space was only 32 which seems to be sufficient to represent the MNIST images. We use a simple ResNet50 network with a single layer at every step that utilizes $3 \times 3$ convolutions. We run our network for 200 epochs in total with a constant learning rate. Better results are obtained if we change $\sigma$, the kernel width, throughout the optimization. We start with $\sigma$ very large, a value of 50, and slowly decrease it, dividing by 2 every 30 steps. The final value of $\sigma$ is 0.78 which yields a rather local support. Convergence curve for our method is plotted in Figure 4. Convergence is generally monotonic and the misfit grows only when we choose $\sigma$, changing the problem to a more challenging one.

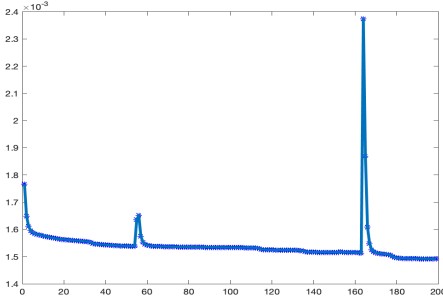

Figure 4: Convergence path of our method. Note the jumps in misfit when we change $\sigma$.

Results of our results are presented in Figure 5. Although not all images look real, we have a very

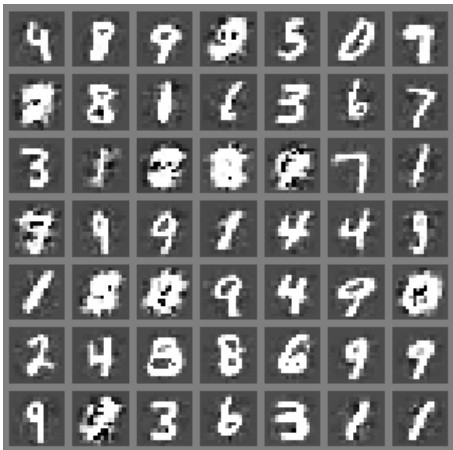

Figure 5: A random set of transformed template vectors that has been trained using encoded MNIST images as reference vectors.

large number (over $80\%$ by visual inspection) that look like they can be taken from the reference set. Unlike the previous experiment where we have a quantitative measure of how successful our approach is, here we have to rely on visual inspection.

## 5 CONCLUSIONS AND FUTURE WORK

In this work we have introduced a new approach for Generative Networks. Rather than viewing the problem as "fooling" an adversary which leads to a minimax we view the problem as a matching problem, where correspondence between points is unknown. This enables us to formulate the problem as strictly a minimization problem, using the theory of optimal mass transport that is designed to match probabilities, coupled with numerical implementation that is based on particles and cloud point registration.

When comparing our approach to typical GANs in low dimensions, where it is possible to construct examples with **known** solution it is evident that our algorithm is superior in terms of iterations to convergence and also in terms of visual inspection. Although we have shown only preliminary results in higher dimensions we believe that our approach is more appropriate for the problem and we will be pursuing variations of this problem in the future. Indeed, is it not better to find a match, that is commonalities, rather than to be adversary?

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
