# OpenReview forum: "FLUID FLOW MASS TRANSPORT FOR GENERATIVE NETWORKS"
_ICLR.cc/2020/Conference — Reject_

### Official Review · AnonReviewer1 · 2019-10-17
**Official Blind Review #1**

**Rating:** 1

**Review:**

This paper proposed a generative network based on fluid flow solutions of mass transport problems. The paper is difficult to follow due to a poor structure and obvious technical mistakes. Detailed comments are as follows:

1. The dual formulation used in the objective of WGANS involves expectations with respect to data distributions. When authors introduced WGANs, it is extremely loose and essential wrong to state that "In the case of WGANs a simpler expression is derived where we use the score directly setting g to the identity and h = −g, and require some extra regularity on the score function".

2. The organisation of the paper makes it difficult to read:
a) While the relevant work is discussed only briefly in Section 1 and contain incorrect statements (see an example above), detailed discussions of two toy examples in the introduction section distract the reading.

b) Section 2 is a combination of related work and proposed work. For example, it starts more like a section introducing the dynamic transport formulation of mass transport problems (Equations 2.3a-2.3c), but in fact it contains the authors' proposed approach (2.4a-2.4b), which makes it difficult to tell the authors' contributions. Moreover, the connection between 2.3a-2.3c to 2.4a-2.4b is not clear to me.

3. Multiple notations are not properly defined or conflicting: what are $\rho(x,t)$ and $\rho(1,x)$?

4. Very limited experimental validation with no comparison to other algorithms.

5. Multiple typos in the paper, e.g. "Equation equation 1.1".

**Experience Assessment:**

I have published one or two papers in this area.

**Review Assessment: Checking Correctness Of Derivations And Theory:**

I assessed the sensibility of the derivations and theory.

**Review Assessment: Checking Correctness Of Experiments:**

I assessed the sensibility of the experiments.

**Review Assessment: Thoroughness In Paper Reading:**

I read the paper at least twice and used my best judgement in assessing the paper.

---

### Official Review · AnonReviewer2 · 2019-10-22
**Official Blind Review #2**

**Rating:** 3

**Review:**

This paper targets training generative adversarial networks with a formulation that is motivated by mass transport of fluid flows. While generalized transport formulations are popular for GANs by now, especially the Earth Mover's distance and the Wasserstein GAN version that this paper is based on, this submission instead frames the problem with a divergence-free flow model inspired by Navier-Stokes. The problem of matching output distributions then becomes one of inferring a suitable flow field that aligns the distributions.

The proposed model is discretized in a Lagrangian manner, and divergence freeness is ensured trivially by a constant particle count. Matching the point clouds in terms of distribution is done via a gaussian mixture model to obtain smooth distributions. (This probably doesn't scale too well to large paricle numbers due to the global support, but the paper also focuses on smaller 2D cases.) As mentioned in the text, the method in the end shares similarities with point cloud registration (ICP).

A synthetic test with a square target shape and an initial centered distribution is shown to highlight that the method can re-distribute parts of the initial cloud towards all sides of the target shape.

As a tougher case, a digit generator with MNIST data is shown, which however, does not reach a level of quality we could expect from other existing methods for generative models.

While I found the initial motivation of the paper quite interesting (and novel as far as I can tell), the discretized version is somewhat disappointing. The smooth matching of point clouds does not retain too much of the initial fluid flow model. Unfortunately, the MNIST test also indicates that the method has problems scaling to higher dimensions. This is a central challenge for GANs, and based on the submission I don't have the impression that the proposed formulation is competitive with existing GAN methods.

**Experience Assessment:**

I have published one or two papers in this area.

**Review Assessment: Checking Correctness Of Derivations And Theory:**

I assessed the sensibility of the derivations and theory.

**Review Assessment: Checking Correctness Of Experiments:**

I carefully checked the experiments.

**Review Assessment: Thoroughness In Paper Reading:**

I read the paper thoroughly.

---

### Official Review · AnonReviewer3 · 2019-10-27
**Official Blind Review #3**

**Rating:** 3

**Review:**

The paper addresses the task of constructing a generative model for data using a novel optimal transport-based method. The paper proposes an alternative view of obtaining generative models by viewing the generation process as a transport problem (specifically, fluid flow mass transport) between two point clouds living in high-dimensional space. To solve the transport problem, a discretization scheme is proposed, which gives rise to a variant of point cloud registration problem, which is solved using numerical optimization. The results are provided on synthetic data and a real MNIST data.

With generative models, and particularly generative adversarial networks (GANs) being notoriously hard to train, alternative ways of constructing generative models are needed. The paper does a good job displaying the potential optimization issues arising when training GANs, and the basic theoretical foundations used in the paper have been validated in prior work. However, with the introduction of a transport-based formulation, its respective issues may arise, that are not described in the paper. Will the optimization always converge, and if yes, to which kind of optimum? What are the requirements for the point clouds R and T?

While the conceptual contribution is that the , the technical novelty is limited and mostly amounts to the appropriate choice of distributions in R and T and applying the discretization schemes to be able to compute the experiments. Unfortunately, the model is not studied theoretically, i.e. no description is given regarding the class of generative tasks that could be solved using the method, or the class of functions that could be learn in such a way.

The experimental evaluation demonstrates on only two simple examples the results of the work. More experiments are needed to fully understand the possibilities of the framework.

The convergence speed is not indicated, and the efficiency of the optimization is not described.

To summarize, I believe the paper should not be accepted in its present (early) form, as (1) more detailed theoretical insight are needed, and (2) much more computational experiments are needed to fully validate the method.

**Experience Assessment:**

I do not know much about this area.

**Review Assessment: Checking Correctness Of Derivations And Theory:**

I assessed the sensibility of the derivations and theory.

**Review Assessment: Checking Correctness Of Experiments:**

I assessed the sensibility of the experiments.

**Review Assessment: Thoroughness In Paper Reading:**

I read the paper at least twice and used my best judgement in assessing the paper.

---

### Decision · Program_Chairs · 2019-12-19

**Decision:**

Reject

**Comment:**

The submission is concerned with providing a transport based formulation for generative modeling in order to avoid the standard max/min optimization challenge of GANs. The authors propose representing the divergence with a fluid flow model, the solution of which can be found by discretizing the space, resulting in an alignment of high dimensional point clouds.

The authors disagreed about the novelty and clarity of the work, but they did agree that the empirical and theoretical support was lacking, and that the paper could be substantially improved through better validation and better results - in particular, the approach struggles with MNIST digit generation compared to other methods.

The recommendation is to not accept the submission at this time.